# Deployable CRISPR-Cas13a diagnostic tools to detect and report Ebola and Lassa virus cases in real-time

Kayla G. Barnes ⓘ et al.#

Recent outbreaks of viral hemorrhagic fevers (VHFs), including Ebola virus disease (EVD) and Lassa fever (LF), highlight the urgent need for sensitive, deployable tests to diagnose these devastating human diseases. Here we develop CRISPR-Cas13a-based (SHERLOCK) diagnostics targeting Ebola virus (EBOV) and Lassa virus (LASV), with both fluorescent and lateral flow readouts. We demonstrate on laboratory and clinical samples the sensitivity of these assays and the capacity of the SHERLOCK platform to handle virus-specific diagnostic challenges. We perform safety testing to demonstrate the efficacy of our HUDSON protocol in heat-inactivating VHF viruses before SHERLOCK testing, eliminating the need for an extraction. We develop a user-friendly protocol and mobile application (HandLens) to report results, facilitating SHERLOCK's use in endemic regions. Finally, we successfully deploy our tests in Sierra Leone and Nigeria in response to recent outbreaks.

#A list of authors and their affiliations appears at the end of the paper.

Ebola virus (EBOV) and Lassa virus (LASV) pose immediate, severe threats to human life and public health, as demonstrated by ongoing outbreaks of EBOV disease (EVD) in the Democratic Republic of the Congo (DRC) and Lassa fever (LF) in Nigeria. Despite their high morbidity and mortality, EVD and LF are difficult to diagnose, because early symptoms, including fever, vomiting, and aches, are often indistinguishable from those of more common tropical diseases[1–3]. Rapid point-of-care diagnostics are vital for facilitating timely clinical care and proper containment[4].

Despite the critical need for rapid point-of-care diagnostics for these viruses, current gold standards lack the logistical feasibility to effectively diagnose cases in endemic regions with limited infrastructure. PCR-based diagnostics are sensitive and can be rapidly developed for emerging or mutating viruses but they are not practical as a point-of-care test, as they require advanced laboratory infrastructure, a cold chain, and expensive reagents. Rapid antigen- and antibody-based tests are deployable but are less sensitive than PCR and take longer to develop; they can also be ineffective in the early/acute stage of infection[5–7], a critical period for supportive care and to contain human-to-human spread.

EBOV and LASV both present distinct diagnostic challenges. The live attenuated rVSVΔG-ZEBOV-GP EBOV vaccine (Merck), currently being deployed to combat the DRC outbreak, produces EBOV GP RNA that can yield false-positive tests by glycoprotein (GP)-targeting assays, including the commonly used GeneXpert reverse-transcriptase quantitative PCR (RT-qPCR)[8–10]; similar false positives will be a concern whenever new live attenuated vaccines are introduced for any virus. In the case of LASV, high genetic diversity in the virus through western Africa hinders the development of diagnostic tools sensitive to all viral strains. The most widely used diagnostic for LF viral detection, a RT-qPCR developed against Josiah strains derived from Sierra Leone (clade IV), has had false-positive and false-negative results when tested against recent clade II samples from Nigeria[11,12].

The recently developed CRISPR-based SHERLOCK (Specific High-sensitivity Enzymatic Reporter unLOCKing) platform provides a promising approach for rapidly adaptable, deployable diagnostics. SHERLOCK utilizes the RNA-targeting protein Cas13a for sensitive and specific detection of viral nucleic acid[13,14]. It pairs isothermal recombinase polymerase amplification (RPA) with crRNA-guided Cas13a detection, which enables specific pairing of Cas13a with the target sequence and signal amplification via Cas13's collateral cleavage activity[13,15,16]. Both amplification and Cas13a-based detection are isothermal, requiring only a low-energy, single-temperature heat block and basic pipette and tips, compatible with point-of-care detection. SHERLOCK can be combined with HUDSON (Heating Unextracted Diagnostic Samples to Obliterate Nucleases), which inactivates pathogens and releases nucleic acid through a combined heat and chemical denaturation, eliminating the need for a column- or bead-based nucleic acid extraction[14]. Our recent work has shown the high sensitivity of SHERLOCK and HUDSON in detecting Zika virus and dengue virus directly from bodily fluids[14], allowing for a fully point-of-care diagnostic. Utilizing this system, we develop a diagnostic test for EBOV and LASV that can be deployed in any setting, requires minimal processing of infectious materials, and accurately reports test results in a user-friendly format.

## Results

**CRISPR-Cas13a diagnostic development and validation for VHFs.** Motivated by the increased severity and frequency of EBOV and LASV outbreaks, we describe here the development and validation of SHERLOCK assays to detect these viruses. The assays can be detected by two readout methods, either fluorescence or lateral flow. The more sensitive fluorescence-based system allowed us to perform extensive validation during the development of our assay, determine the length of amplification time needed for viral detection, and determine the limit of detection (LOD). The lateral flow readout, which we then validated further, utilizes a commercially available detection strip to provide semi-quantitative point-of-care detection of the virus.

We developed a SHERLOCK EBOV assay to target the *L* gene of the EBOV Zaire strain, which accounts for the majority of known clinical cases of EBOV infections, including the two largest and most recent EVD epidemics[17,18]. We used primer design applications (CATCH[19]) to identify an optimal target within a conserved region of the *L* gene, thus avoiding potential false-positive results caused by the rVSVΔG-ZEBOV-GP EBOV vaccine (Fig. 1a and Supplementary Tables 1–3). Our assay detected synthetic DNA at concentrations as low as 10 copies/µl using either fluorescent or lateral flow readout (Fig. 1b, c). As EBOV, LASV, and Marburg virus (MARV) infections present with similar symptoms and have been known to co-circulate, we tested for cross-reactivity using seedstock and synthetic DNA of each virus; our assay showed no cross-reactivity to either LASV or MARV (Fig. 1d).

We validated the SHERLOCK EBOV assay at the Broad Institute using 16 clinical samples taken from suspected EVD patients in Sierra Leone during the 2014–2016 West Africa outbreak. For safety reasons, we tested complementary DNA (which is not infectious) and benchmarked the results against previously generated sequencing data[17,20] (Fig. 1e). Of the 16 samples, 12 were positive for EBOV by sequencing, all 12 of which were positive by SHERLOCK. The four sequencing-negative samples were negative by SHERLOCK (100% sensitivity, 100% concordance).

We also developed and validated SHERLOCK assays for LASV, a challenging target because of the virus's extreme genetic diversity, both within and especially between clades[21]. Currently, two clades—clade II, localized in Nigeria, and clade IV, localized in Sierra Leone[22]—account for over 90% of known clinical infections[21,22]. Given this extreme genetic diversity, we designed two LASV SHERLOCK assays (Supplementary Tables 1–3). The first assay (LASV-II) targets clade II. To ensure detection of all known genomes in this highly divergent clade, the assay contains two multiplexed crRNAs (LASV-IIA and LASV-IIB) (Fig. 2a). When we compared the two LASV-II crRNAs to an alternative assay with only one crRNA, the former detected LASV more quickly and identified an additional positive sample (Fig. 2b). The second assay (LASV-IV) targets clade IV; in this clade, we found a more conserved region that enabled us to use a single crRNA. The LASV-II assay was sensitive down to 10 copies/µl with a fluorescent readout (Supplementary Fig. 1a) and 100 copies/µl using lateral flow strips (Supplementary Fig. 1c); the LOD for LASV-IV was 100 copies/µl for the fluorescence-based assay (Supplementary Fig. 1b) and 1000 copies/µl for the lateral flow assay (Supplementary Fig. 1d). Neither the LASV-II nor the LASV-IV assay cross-reacted with synthetic DNA from the other clade or with known positive patient samples from the other clade's geographic region, nor did they cross-react with EBOV or Marburg seedstock cDNA (Fig. 2c, d). The LASV SHERLOCK assays are thus both species and clade specific, and therefore region specific, which can help distinguish possible imported cases from local transmission.

**Deployment of CRISPR-Cas13a diagnostics.** Next, we deployed the EBOV assay to our collaborators in Sierra Leone to test

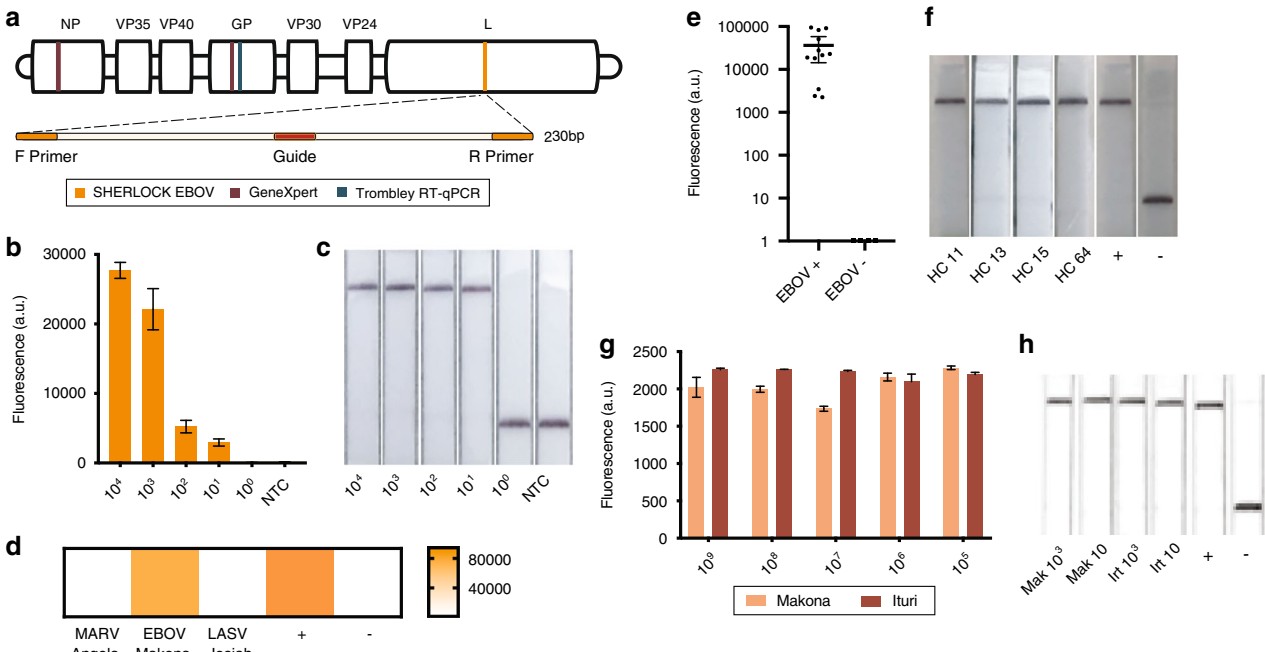

**Fig. 1 Detection of EBOV. a** Schematic of the SHERLOCK EBOV assay. **b**, **c** Detection of a serial dilution of EBOV synthetic DNA using (**b**) mean fluorescence of three technical replicates and (**c**) lateral flow readouts. Error bars indicate ±1 SD for three technical replicates. **d** Test of cross-reactivity using MARV, EBOV, and LASV viral seedstock cDNA. Heat map is measured in Fluorescence (a.u.). **e** SHERLOCK testing of cDNA extracted from 12 confirmed EBOV-positive and 4 confirmed EBOV-negative samples collected from suspected EVD patients during the 2014 outbreak in Sierra Leone. Error bars indicate 95% confidence interval. **f** Four of the samples from **e** were also tested by collaborators using lateral flow detection. **g**, **h** Detection of serial dilution of synthetic RNA from Ituri, DRC and Makona, Sierra Leone using (**g**) fluorescence where error bars indicate ±1 SD for three technical replicates and (**h**) lateral flow readouts carried out at USAMRIID. Source data are in the Source Data file.

patient samples stored from the 2014–2016 outbreak using the point-of-care lateral flow assay. This allowed us to validate and assess the practicality of the SHERLOCK assay in a setting with previously circulating EBOV and limited infrastructure. As a head-to-head comparison, we identified 4 whole blood (WB) in trizol aliquots from the same patients tested in our first panel of 16 samples. These samples were stored at the Kenema Government Hospital (KGH) biobank under variable temperature conditions (−20 °C with multiple power cuts). Using the same protocol as for the panel of 16 (Supplementary Fig. 2), all 4 samples were positive by SHERLOCK, consistent with the results obtained at the Broad Institute (Fig. 1f), despite multiple years of an imperfect cold chain.

The SHERLOCK EBOV assay was also highly efficient at detecting a more recent EBOV variant from the DRC. We tested a synthetic version of a 2018 Ebola isolate from Ituri Province (Ituri isolate 18FHV089), DRC (Fig. 1g, h), and, as validation, a 2014 Makona isolate from Sierra Leone that underwent the same synthetic generation process (see "Methods")[23,24]. Utilizing both the SHERLOCK fluorescence-based and lateral flow-based assays, the Makona and Ituri isolates were both detected at levels down to 10 copies/μl. The DRC isolate was genetically distinct from the 2014 isolate but maintained the key conserved stretch on the L gene that the SHERLOCK assay targets, which remains conserved on all available genomes from the ongoing DRC outbreak.

We also tested our LASV assay in Sierra Leone and Nigeria using clinical samples. We evaluated the sensitivity on a panel of ten RNA and cDNA samples per clade, derived from suspected LF patients (Fig. 2e, g). We compared SHERLOCK results using both fluorescent and lateral flow readouts head-to-head with the Nikisin RT-qPCR assay[25] and benchmarked both results against sequencing data (Fig. 2f, h). The LASV-II fluorescent readout was positive for all seven sequencing-positive samples and negative

for all three sequencing-negative samples (100% sensitivity, 100% concordance), as was the Nikisin RT-qPCR. The LASV-II lateral flow readout failed to detect one sequencing-positive sample. For the LASV-IV assay, SHERLOCK performed significantly better than RT-qPCR. SHERLOCK was again positive for all seven sequencing-positive samples (100% sensitivity), whereas the Nikisin assay was only 40% sensitive and our in-house Broad RT-qPCR assay (Supplementary Tables 4 and 5), developed on all recent LASV genomes, was only 50% sensitive. The three sequencing-negative samples were negative by SHERLOCK (100% concordance). The low detection rate of clade IV by RT-qPCRs is likely due to multiple mismatched base pairs where the primers anneal; despite this, Nikisin continues to be a primary diagnostic.

**Safety analysis and efficacy of heat inactivation.** Reducing exposure to viral hemorrhagic fevers (VHFs) among healthcare workers is critical for the safe and effective use of diagnostic tests. For diagnostic test design, this requires ensuring that the sample is fully inactivated and, where possible, using non-invasive sample types. To this end, we combined the SHERLOCK assay with the HUDSON technique, which integrates heat inactivation with TCEP : EDTA to denature RNAses and release nucleic acid from viral particles, thus eliminating the need for RNA extraction. Furthermore, as LASV and EBOV are secreted in saliva and urine[26,27], HUDSON enables disease diagnosis without the need for an invasive blood draw or specialized equipment, resulting in a faster end-to-end processing time. To confirm HUDSON's efficacy for viral inactivation and to determine the most sensitive HUDSON protocol for SHERLOCK use, we carried out tests at the BSL4 laboratory facility at the NIH Integrated Research Facilities.

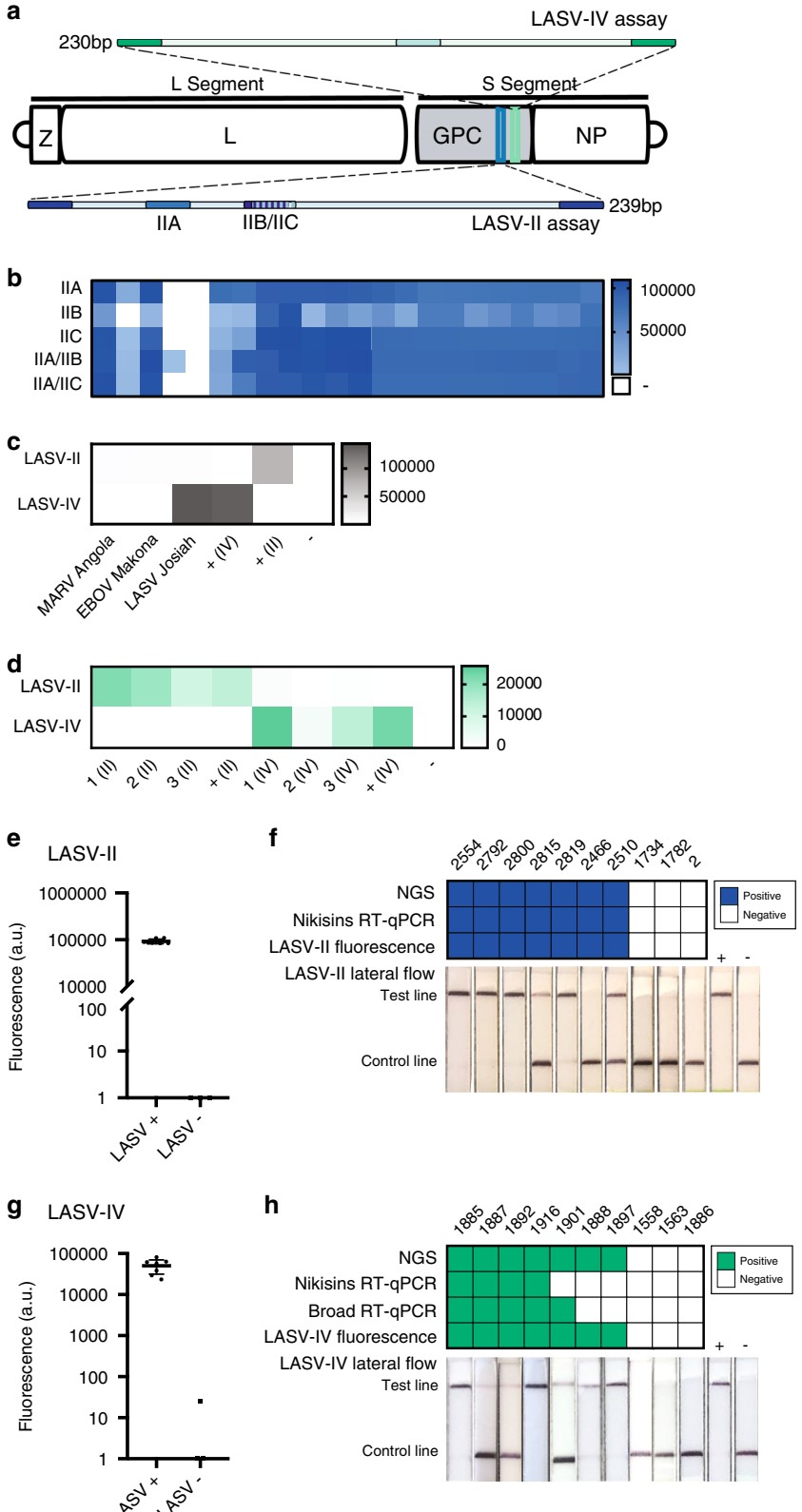

We first showed that HUDSON successfully rendered viruses inactive in three sample types. We spiked human WB, urine, and saliva with the live EBOV Mayinga variant and confirmed that samples had viral activity using an initial plaque assay (Fig. 3a). Samples underwent serial dilution to mimic variation in viral load and were then heat and chemical treated using HUDSON.

We performed HUDSON using two conditions, either 95 °C for 10 min or 70 °C for 30 min, to determine the most effective heat-inactivation protocol. We used a standard plaque assay, two passages in Vero cells, to determine presence or absence of replication-competent virus. After HUDSON treatment, no viable virus was detected at either temperature, showing complete inactivation at all

**Fig. 2 Detection of LASV clade II and IV. a** Schematic of LASV SHERLOCK assays targeting the two most common clades of LASV: clades II (LASV-II assay) and IV (LASV-IV assay). For the LASV-II assay, three crRNAs were designed and tested. Two crRNAs are multiplexed to encompass the clade's genetic diversity (IIA/IIB or IIA/IIC). Each crRNA was tested using three technical replicates. **b–d** Heat maps are measured in fluorescence (a.u.). **b** Detection of LASV RNA from suspected LF clinical samples using crRNAs IIA, IIB, IIC, or a combination of crRNAs. **c** Test of cross-reactivity between different viral species using MARV, EBOV, and LASV viral seedstock cDNA. The LASV-II and LASV-IV assays do not cross-react with MARV or EBOV seed stocks. **d** Test of cross-reactivity between LASV clade-specific assays using clinical samples from recent outbreaks in Nigeria and Sierra Leone. The LASV-II and LASV-IV assays provide clade-specific detection. **e** SHERLOCK testing using the LASV-II assay of RNA extracted from seven confirmed LASV-positive and three confirmed LASV-negative samples collected from suspected LF patients in Nigeria during the 2018 outbreak. Error bar indicates 95% confidence interval. **f** Results from **e** were compared head-to-head to those from the gold standard Nikisins RT-qPCR assay, next-generation sequencing (genome assembled), and lateral flow detection. **g** SHERLOCK testing using the LASV-IV assay of RNA extracted from seven confirmed LASV-positive and three confirmed LASV-negative samples collected from suspected LF patients in Sierra Leone. Error bar indicates 95% confidence interval. **h** Results from **g** were compared head-to-head to those from the gold standard Nikisins RT-qPCR assay, a second Broad RT-qPCR, NGS, and lateral flow detection. Source dare are in the Source Data file.

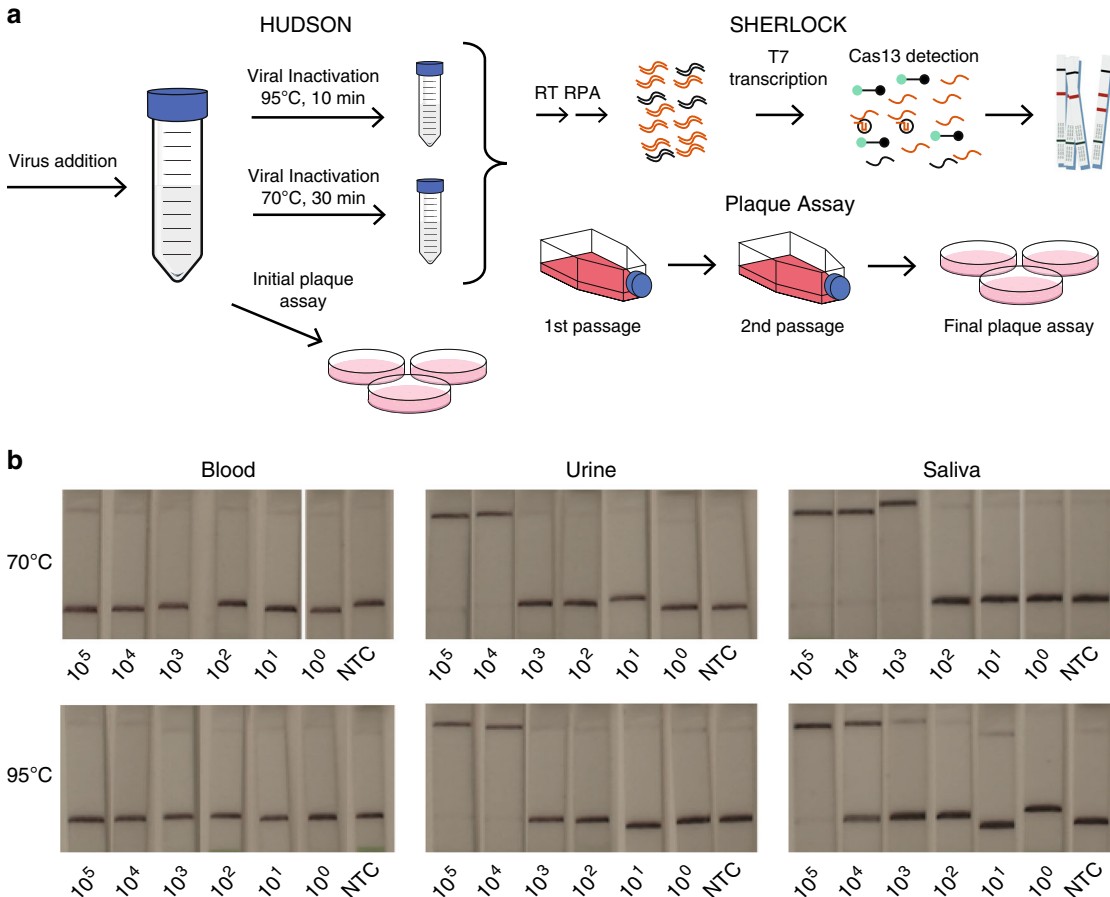

**Fig. 3 HUDSON safety testing. a** Schematic overview of the HUDSON, SHERLOCK inactivation validation. Viral inactivation includes dilution with EDTA : TCEP and a 20 min 37 °C inactivation of nucleases. All final results were determined using lateral flow due to the inability to carry out appropriate fluorescent analysis in the BSL4 facility. **b** Lateral flow detection of spiked blood, urine, and saliva inactivated at either 70 °C or 95 °C. Serial dilution shown are PFU/mL. All assays were carried out in the BSL4 facility.

concentrations and confirming the safety of the HUDSON-SHERLOCK platform. To ensure safety, clinical and laboratory staff should use appropriate personal protective equipment or a glove box until a sample is fully inactivated.

We then performed SHERLOCK on the HUDSON-inactivated samples to establish how HUDSON temperature conditions affect SHERLOCK's performance. The serially diluted EBOV samples were tested by SHERLOCK using the lateral flow readout (Fig. 3b) and the results were compared to the GeneXpert diagnostic. Both heat-inactivation conditions performed equally. Using our combined HUDSON-SHERLOCK method, we detected virus down to 1.1E + 05 PFU/mL in WB, 1.2E + 05 PFU/mL in urine,

and 9.4E + 03 PFU/mL in saliva (Supplementary Table 6). When considering cycle threshold ≤ 36 as definitive positives[28], GeneXpert was more sensitive than SHERLOCK for WB (4.4E + 02 PFU/mL) and urine (1.1E + 02 PFU/mL), but comparable for saliva (1.1E + 03 PFU/mL but for only NP detection, 9.4E + 03 PFU/mL for GP detection). Ultimately, our HUDSON testing highlights the potential to safely and sensitively test saliva in suspected patients, minimizing the need for more invasive blood draws and increasing safety for healthcare workers.

**HandLens: a mobile application for diagnostic analysis.** The lateral flow readout of the current SHERLOCK protocol can be

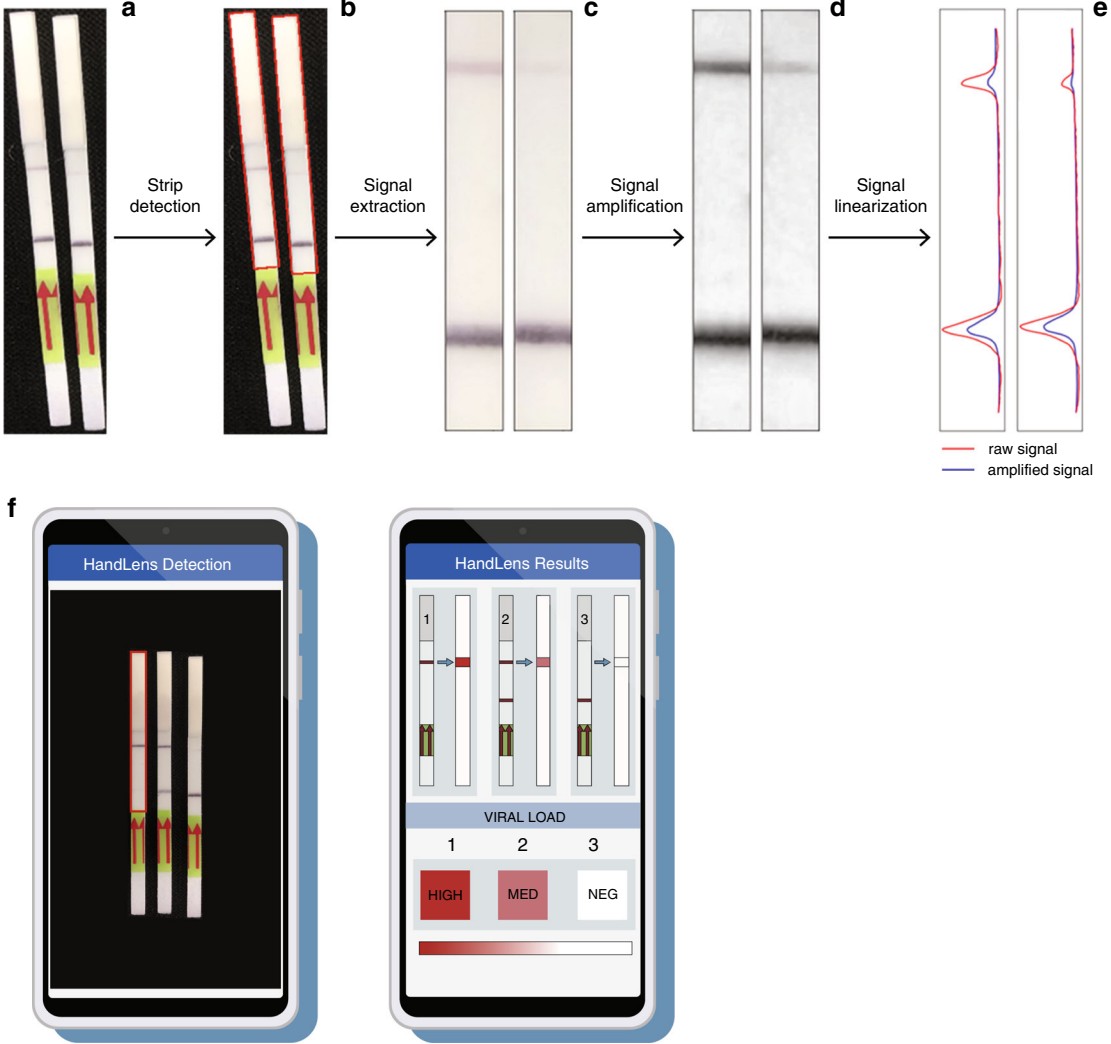

**Fig. 4 Quantification of SHERLOCK lateral flow strips using HandLens, an Android app prototype.** Internal image analysis pipeline of the SHERLOCK detector app (HandLens). **a** Images of two positive sample lateral flow strips are imported to the app. **b** The relevant signal regions of the lateral flow strips are detected and demarcated by red bounding boxes. **c** Bilateral filtering is used to extract and smoothen the signal regions from the raw input image. **d** Contrast within the image is increased by applying contrast limited adaptive histogram equalization (CLAHE). **e** The signal is linearized for downstream signal processing; the red curves indicate the signal extracted after applying CLAHE, whereas the blue curves indicate the signal levels if the CLAHE step is skipped. **f** The strip reader app works by allowing the user to take a picture of the test strips where a rectangle can be used to select the control strip on the leftmost side. The raw image data is sent to a backend server that runs the signal detection algorithm and returns the binary and semi-quantitative predictions for each strip.

difficult to interpret for low concentration samples due to the correlation of band darkness with viral load. Critical for a deployable rapid diagnostic and surveillance tool is an easy-to-use interface that produces and reports a consistent readout free of operator bias. In addition, lateral flow band strength has the potential to generate a semi-quantitative result. To exploit that potential and to facilitate accurate readout, we developed a mobile phone application called HandLens that captures (Fig. 4a, b) and analyzes (Fig. 4c–e) an image of one or more lateral flow strips to quantify test results and resolve ambiguous readouts (Fig. 4). Using a prototype version of the HandLens app, we tested a dilution series (ranging from $10^5$ to 10 copies/µl) from four EBOV samples and compared our results to RT-qPCR. This yielded estimates of 93% accuracy, 91% sensitivity, and 100% specificity from a total of 21 strips (Supplementary Fig. 3a). This app can be adapted for use on any smartphone or tablet, allowing a clear, unbiased diagnostic readout.

## Discussion

The recent severe acute respiratory syndrome coronavirus 2 (SARS-CoV-2) pandemic has highlighted the need for rapid deployable diagnostics that decrease healthcare worker exposure. Although there has been great improvement with platforms like Abbot and GeneXpert, these RT-qPCRs still require expensive machines and laboratory infrastructure. Furthermore, this pandemic and other recent viral outbreaks have demonstrated the lag time it takes to create a sensitive antibody- or antigen-based rapid diagnostic test (RDT). SHERLOCK provides an alternative viral diagnostic method that addresses these shortcomings. In summary, we have developed sensitive, specific, point-of-care CRISPR-based diagnostics for EBOV and LASV, two hemorrhagic fever viruses that pose immediate global threats. We have validated these diagnostics on laboratory and patient samples, including deployment for testing in partner laboratories in Sierra Leone and Nigeria. In addition, we have shown that the HUDSON protocol

not only removes the need for extraction but also inactivates EBOV to allow for a safe low-tech test, and we demonstrated that non-invasive samples including saliva and urine can be used for rapid detection, eliminating the need for a blood draw and increasing safety for clinical staff testing for suspected VHF. We provide a user-friendly readout that can be documented using a mobile device to allow for greater reproducibility and immediate reporting. The HUDSON-SHERLOCK assay minimizes testing time and handling of infectious samples, reduces the cost to less than $1 USD[29] per sample, and can be run with minimal equipment using only solar power or a small generator to allow for quick diagnostics in any environment, showcasing the growing capabilities of CRISPR-based diagnostics for viral detection.

## Methods

**Ethical approval for the use of clinical samples**. All patient samples used for this study were de-identified and were obtained through studies that were evaluated and approved by the institutional review boards at the Irrua Specialist Teaching Hospital (Irrua, Nigeria), Redeemer's University (Nigeria), KGH (Sierra Leone), Sierra Leone Ministry of Health, Ministry of Health of the DRC, and Harvard University (Cambridge, Massachusetts).

LF patients were recruited for this study using protocols approved by human subjects committees at Harvard University, Broad Institute, Irrua Specialist Teaching Hospital, KGH, Oyo State Ministry of Health, Ibadan, Nigeria, and Sierra Leone Ministry of Health. All patients were treated with a similar standard of care and were offered the drug Ribavirin, whether or not they decided to participate in the study.

Due to the severe outbreak for EVD, patients could not be consented through our standard protocols. Instead use of clinical excess samples from EVD patients was evaluated and approved by Institutional Review Boards in Sierra Leone and at Harvard University. The Office of the Sierra Leone Ethics and Scientific Review Committee, the Sierra Leone Ministry of Health and Sanitation, and the Harvard Committee on the Use of Human Subjects has granted a waiver of consent to use de-identified samples collected from all suspected EVD patients receiving care during the outbreak response. The Sierra Leone Ministry of Health and Sanitation also approved shipments of non-infectious non-biological samples from Sierra Leone to the Broad Institute and Harvard University for genomic studies of outbreak samples.

Protocol Title: Genomic Characterization and Surveillance of Microbial Threats in West Africa
Principal Investigator: Pardis C. Sabeti
Protocol #: IRB19-0023
Funding Source: The Broad Institute-5700161-5500000755 (Active), NIH; Military HIV Research Program and Henry M. Jackson Foundation, NIH; Bill and Melinda Gates Foundation
Protocol Title: Sierra Leone Lassa and Ebola Case Control (includes Ebola Clinical Excess from Deceased Patients), Nigeria Lassa Case Control
Protocol #: CUHS 21288 and ORSP 2202
Harvard University
Funding Source: The Broad Institute-5700161-5500000755 (Active), NIH; Military HIV Research Program and Henry M. Jackson Foundation, NIH; Bill and Melinda Gates Foundation

**Sample preparation**. Patient samples were taken by clinical staff using appropriate personal protective equipment. Inactivation of samples occurred in their country of origin. Samples were then shipped to the Broad Institute or tested at the local center (Nigeria, Sierra Leone). Samples were inactivated in AVL buffer (Qiagen) or TRIzol (Life Technologies) following standard operating procedures. Samples were stored in liquid nitrogen or at −20 °C. RNA was isolated at the clinical site using the QIAamp Viral RNA Minikit (Qiagen) according to the manufacturer's protocol. Poly(rA) and host rRNA were depleted using RNase H selective depletion, using 616 ng oligo (dT) (40 nt long) and/or 1000 ng DNA probes complementary to human rRNA. Samples then underwent RNase-free DNase using a kit (Qiagen) according to the manufacturer's protocol. AMPure RNA clean beads (Beckman Coulter Genomics) were used to clean and concentrate samples. cDNA synthesis was performed using the Superscript III kit (Thermo Fischer) plus dNTPs, random primers, and SUPERASE-IN for first-strand synthesis. Then, the 10× second-strand buffer kit (New England Biolabs), plus *Escherichia coli* DNA ligase, *E. coli* DNA polymerase, Rnase H, and dNTPs were used for second-strand synthesis. Samples then underwent a final AMpure DNA beads clean-up[30].

**SHERLOCK assay design**. To design RPA primers and crRNAs, we identified conserved regions of the EBOV and LASV genomes. For the EBOV assay, we used an alignment based on all published sequences. The highly conserved areas of the EBOV genome allowed us to design numerous efficient crRNAs, with two targeting the NP-gene and two targeting the *L* gene (Supplementary Fig. 4a), neither of

which is expected to cross-react with the live attenuated vaccine. We developed an EBOV assay using the most efficient crRNA based on peak fluorescence and minimum time required for SHERLOCK detected to reach saturation (Supplementary Fig. 4b). For the LASV-IV assay, we used an alignment based on all published LASV clade IV sequences (Sierra Leone)[21]. For the LASV-II assay, we used an alignment based on all published LASV clade II sequences (Nigeria)[21,22]. We then used a 50 bp sliding window to identify flanking conserved areas. We identified 21–29 bp primers and a 29 bp crRNA for the LASV-IV assay (Fig. 2a). Due to the high diversity in clade II, one crRNA, even with up to six degenerate bases, did not encompass all known genomes. We identified three crRNAs within the same 200 bp region and tested these in tandem (Fig. 2b).

**RPA reactions**. All RPA reactions were carried out using the Twist-Dx RT-RPA kit according to the manufacturer's instructions. All reactions were run for 20 min. Primer concentrations were 480 nM. For reactions with RNA input, Murine RNase inhibitor (NEB M3014L) was added at a concentration of 2 units/μl. For a complete list of RPA primer names and sequences, see Supplementary Table 1.

**Production of LwaCas13a and crRNAs**. LwaCas13a was purified by Genscript. The crRNAs were determined by aligning all known genomes and using our CATCH method[19] to identify conserved areas on a sliding scale and were synthesized by Integrated DNA Technologies.

**CAS13a detection reactions**. For detection reaction, refer to our one-page user-friendly protocol (Supplementary Fig. 2). Detection assays were performed for either plate reader (fluorescent) or lateral flow detection. Broadly, Cas13a, crRNAs, T7 polymerase (New England Biolabs), RNasae inhibitors (New England Biolabs), buffer (CB—40 mM Tris-HCl, 60 mM NaCl, 6 mM MgCl₂, pH 7.3), MgCl₂ (rNTPs (New England Biolabs), and either a fluorescent substrate reporter (RNase alert v2) or LP probe (Tewist-Dx) were combined. Detection mix was combined with the RPA reaction and incubator for 1 h at 37 °C[13,14]. For multiplexed crRNAs in LASV-II assays, the total volume of crRNA was doubled. Reactions were run on a Biotek Cytation 5 multi-mode reader. All reactions were run in triplicate alongside a no-template control. Fluorescence kinetics were measured via a monochrometer with excitation at 485 nm and emission at 520 nm, with a reading every 5 min. EBOV assays were run for 1 h and LASV assays were run for 3 h. Reported fluorescence values are specified as background-subtracted or template-specific. For EBOV detection, the crRNA EBOVA was used unless otherwise noted. For detection of LASV clade IV, the crRNA LASV-IVA was used. For detection of LASV-II, an equal mix of crRNAs LASV-IIA and LASV-IIB was used. For a complete list of crRNA names and sequences, see Supplementary Table 2.

**Lateral flow detection reactions**. Lateral flow detection reactions were performed as described using commercially available detection strips according to manufacturer's instructions (Milenia Hybridetect 1, Twist-Dx, Cambridge, UK).

**Data analysis**. For all fluorescence values, background-subtracted fluorescence was calculated by subtracting the minimum fluorescence value, which occurred between 0–20 min, from the final fluorescence value. For all fluorescence values reported for patient samples, found in Figs. 1e, g and 2b, e, h, target-specific fluorescence was calculated by subtracting the mean background-subtracted fluorescence of the no template control from the mean background-subtracted fluorescence of a given target with the same crRNA at the same time point.

**LOD experiments**. To determine the sensitivity of SHERLOCK assays, assay-specific synthetically derived DNA templates were derived from clade- and virus-specific alignments[1,17,20–22]. Synthetically derived DNA (≤500 bp and non-replicant competent) templates were used as input into the RPA reaction at concentrations from 10⁴ copies/μl to 1 copy/μl with a 1 : 10 dilution series. Each crRNA was also tested on a no-input negative control. Reactions were run twice, using both the fluorescent readout and the lateral flow readout. All reactions were run in triplicate for the fluorescent readout.

**Cross-reactivity experiments**. To assess the cross-reactivity of assays with other viruses known to cause hemorrhagic symptoms, all assays were tested on LASV (Josiah), EBOV (Makona), and MARV (Angola) viral cDNA seed stocks. Each assay was also tested on a positive control containing an assay-specific synthetically derived DNA template at a concentration of 10⁴ copies/μl and on no-input negative control. Synthetically derived DNA were short fragments that encompassed the primers and around 20 base pairs on both the 5′- and 3′-end. All fragments were non-replicant competent. All reactions were run in triplicate. The LASV assays were also assessed for clade-specific detection. The LASV-IV assay was tested on three RNA patient samples from clade II and the LASV-II assay was tested on three RNA patient samples from clade IV. Each assay was also tested on a positive control containing an assay-specific synthetically derived DNA template at a concentration of 10⁴ copies/μl and on no-input negative control. All reactions were run in triplicate.

**Validation on patient samples**. The LASV-IV assay was validated on a panel of 12 RNA samples collected from suspected LF patients in Sierra Leone[21,22]. Seven of these samples were confirmed LASV positive by antigen-based RDT, enzyme-linked immunosorbent assay (ELISA) IgM, and RT-qPCR[31]; four were confirmed LASV negative by RDT, ELISA, and RT-qPCR; one did not have the full panel of tests. The assay was also tested on a positive control containing synthetically derived cDNA and a no-input control. All reactions were run in triplicate. A subset of these samples was tested using the lateral flow readout. Results were compared to RT-qPCR results and sequencing results. The LASV-II assay was validated on a panel of 12 cDNA samples collected from suspected LF patients in Nigeria during the 2018 outbreak[22]. Nine of these samples were confirmed LASV positive and three were confirmed LASV negative. The assay was also tested on a positive control containing synthetically derived cDNA at a concentration of $10^4$ cp/µl and a no-input control. All reactions were run in triplicate. A subset of these samples was also tested using the lateral flow readout. Results were compared to RT-qPCR results and sequencing results.

The EBOV assay was validated on a panel of 16 cDNA and RNA samples collected from suspected EVD patients in Sierra Leone during the 2014 outbreak[17,20]. Twelve of these samples were confirmed EBOV positive and four were confirmed EBOV negative. The assay was also tested on a positive control containing synthetically derived cDNA at a concentration of $10^4$ cp/µl and a no-input control. All reactions were run in triplicate. To validate the lateral flow readout, the EBOV assay was tested on four EBOV-positive cDNA samples, alongside a positive and a no-input control.

**RT-qPCR experiments**. RT-qPCR for LASV detection was performed using the Power SYBR Green RNA-to-Ct 1-step RT-qPCR kit (Thermo Fisher) according to the manufacturer's instructions. Reactions were performed on a LightCycler 96 machine (Roche). For detection of LASV clade II, the primers Nikisins_F and Nikisins_R were used[25]. For detection of LASV clade IV, the in-house assay, including primers Broad_F and Broad_R, was also used with the TaqMan RNA-to-CT 1-Step Kit (Applied Biosystems). For a complete list of primer names and sequences, see Supplementary Table 4; for a complete list of probe names and sequences, see Supplementary Table 5.

**EBOV DRC experiments**. All samples tested at USAMRIID underwent SHER-LOCK as described above with the exception of the fluorescent readout which was conducted on a BioRad CFX96. The genomes of both the Makona and Ituri isolates were sequenced with Illumina technology (MiSeq for Makona and iSeq for Ituri) in-country as described in ref. [24]. Briefly, once the genome sequences were obtained, sub-genomic fragments were commercially synthesized and then assembled into plasmids encoding full genomes at USAMRIID. The RNAs were in vitro transcribed from the full genome plasmids. All SHERLOCK work on these isolates was carried out at USAMRIID.

**Safety testing of HUDSON and SHERLOCK**. Before inactivation all samples should be treated using BSL4 safety conditions. All clinical and laboratory staff should wear full PPE and/or use a glove box to acquire and handle patient samples. We also recommend that all equipment should be decontaminated before and after each use. Samples remain highly infectious until the full heat inactivation protocol can be carried out.

Viral titers for each sample were determined by plaque assay; a six-well plate with a confluent monolayer of VeroE6 cells was infected with a predetermined volume of sample. The wells are then overlaid with a medium to ensure the monolayer health and incubated for a set amount of time. If there is live virus in the sample, this virus will infect and kill a cell, and spread cell-to-cell creating a plaque, or a clearing of cells. Plaques are then visualized by a crystal violet stain and counted to determine viral titer. Samples underwent a serial dilution and then were heat-inactivated using methods described in[14]. Briefly, a 1 : 100 solution of 0.5 M EDTA to TCEP (Thermo) was used to decrease RNase degradation. The EDTA : TCEP was added to spiked samples at a ratio of one part EDTA : TCEP to four parts samples. First, samples were heated to 37 °C for 20 min to inactivate nucleases. The samples were then heated to 95 °C for 10 min or 70 °C for 30 min as described, followed by the SHERLOCK assay as described above. Only Eppendorph Safe-Lock tubes or cryovials with a screw top should be used for heat inactivation and the outside of the tubes should be decontaminated before and after heat inactivation.

The samples then underwent primary passaging. Samples were added to a T25 flask, incubated for 1 h at 37 °C at 5% $CO_2$, then replenished with media and incubated for 7 days. Seven days post infection (dpi) pictures were taken of each flask to assess for cytopathic effect. Secondary passage was performed to assess for any residual virus particles not detected in the primary passage. All media was transferred from the T25 flask to a T75 flask incubated for 1 h at 37 °C at 5% $CO_2$, replenished with media and incubated for 7 dpi, and then photographed. Following the primary and secondary passaging, a final plaque assay was performed to determine viral titer and to assess viral clearance or reduction. For diagnostic comparison, qPCR samples were run on the Cepheid GeneXpert platform. Sample were inactivated in Xpert lysis buffer at room temperature for 10 min and then run with an Xpert Ebola assay cartridge[10].

**HandLens: the lateral flow reader app**. First, the signal-containing section of the lateral flow strip is detected using OpenCV's[32] contour detection routines. This region is extracted and transformed into a smooth two-dimensional image using bilateral filtering. The result is enhanced using contrast limited adaptive histogram equalization[33], which has the property of increasing signal contrast in local regions. This signal is then linearized by integrating pixel intensity over each row. Finally, a signal is marked as positive for viral load if the signal intensity in the test band of the strip is above a certain user-defined threshold compared to the control strip, and negative if the signal intensity is too low. We also developed quantifiable signal graphs of each band and controlled for shadows and image contrast by applying a contrast-improvement algorithm (Supplementary Fig. 3b).

**Reporting summary**. Further information on research design is available in the Nature Research Reporting Summary linked to this article.

## Data availability

All data presented in this manuscript and generated for this work are included in the written text and figures including a list of all RPA primer sequences (Supplementary Table 1), crRNA spacer sequences (Supplementary Table 2), RT-qPCR primer sequences (Supplementary Table 4), and RT-qPCR probe sequences (Supplementary Table 5). All results generated using a fluorescence readout are the average of three replicates. Any other relevant data are available from the authors upon reasonable request. Source data are provided with this paper.

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

## Acknowledgements

Opinions, interpretations, conclusions, and recommendations are those of the authors and are not necessarily endorsed by the United States Army or the National Institute of Health. We thank the staff of the Joint West Africa Research Group, the Walter Reed Army Institute of Research, the Henry M. Jackson Foundation for the Advancement of Military Medicine, and the U.S. Embassy in Abuja, Nigeria, for their collaborative support and our colleagues at the Irrua Specialist Teaching Hospital, Nigeria, and Kenema Government Hospital, Sierra Leone. We thank Dr. Christopher Moxon for his critical edits of the manuscript. This work is supported by grants from the National Institute of Allergy and Infectious Diseases (NIAID), National Institutes of Health (NIH) (U19AI110818 and R01AI114855 to the Broad Institute, P.C.S.), Henry M Jackson Foundation (W81XWH-18-2-0040 to P.C.S.), DARPA (D18AC00006 to the Broad Institute, P.C.S.), the Bill and Melinda Gates Foundation (OPP1192035 to Harvard University, P.C.S.), NIH-Fogarty (K01TW010853 to HTHCSPH, K.G.B.), and P.C.S. is supported by the Howard Hughes Medical Institute.

## Author contributions

K.G.B., A.E.L., and A.N. conceived of the study. P.C.S., C.T.H., C.M., and C.A.F. oversaw the development of the SHERLOCK assays. R.F.G., D.J.P., and S.F.S. provided input on study design and analysis. H.C.M., L.B., M.B., and B.S. provided validation data for the SHERLOCK assay. A.C. and S.S. developed the mobile application with help from A.E.L. R.G., B.D.K., and L.E.H. performed and oversaw the BSL4- safety validation. B.B. and G.P. generated and performed the diagnostics on the DRC samples. SHERLOCK laboratory experiments were performed by K.G.B., A.E.L., A.N., A.E.L., A.C., C.K.B., J.U., F.A., T.O., and M.K. Patient recruitment and care and data analysis of patient samples was carried out by K.G.B., A.E.L., K.J.S., S.B.M., J.U., F.A., T.O., I.O., J.D.S., M.M., M.F.A., A.B.T., Z.F.P., M.I., D.S.G., and K.M. All authors contributed to the drafting and editing of this manuscript.

## Competing interests

K.G.B., A.E.L., C.A.F., P.C.S., and C.M. are inventors on patent PCT/US2019/054561 held by the Broad Institute and related to this work. The patent covers all primers, crRNAs, and SHERLOCK technology. P.C.S. is a co-founder of, shareholder in, and advisor to Sherlock Biosciences, Inc., as well as a Board member of and shareholder in Danaher Corporation.

## Additional information

Kayla G. Barnes [1,2,3,25] ✉, Anna E. Lachenauer [1,4,25] ✉, Adam Nitido [5,6], Sameed Siddiqui [1,7], Robin Gross [8], Brett Beitzel [9], Katherine J. Siddle [1,10], Catherine A. Freije [1,6], Bonnie Dighero-Kemp [8], Samar B. Mehta [1,11], Amber Carter [1], Jessica Uwanibe [12,13], Fehintola Ajogbasile [12,13], Testimony Olumade [12,13], Ikponmwosa Odia [14], John Demby Sandi [12,15], Mambu Momoh [12,15], Hayden C. Metsky [1,16], Chloe K. Boehm [1], Aaron E. Lin [1,6], Molly Kemball [1,10], Daniel J. Park [1], Luis Branco [17], Matt Boisen [17], Brian Sullivan [18], Mihret F. Amare [19,20], Abdulwasiu B. Tiamiyu [19,21], Zahra F. Parker [19,20], Michael Iroezindu [19,21], Donald S. Grant [15,22], Kayvon Modjarrad [19], Cameron Myhrvold [1,10], Robert F. Garry [17,23], Gustavo Palacios [9], Lisa E. Hensley [8], Stephen F. Schaffner [1,2,10], Christian T. Happi [2,12,13,14], Andres Colubri [1,10] & Pardis C. Sabeti [1,2,10,24]

[1]Broad Institute of MIT and Harvard, Cambridge, Massachusetts, USA. [2]Department of Immunology and Infectious Diseases, Harvard T.H. Chan School of Public Health, Harvard University, Boston, Massachusetts, USA. [3]MRC-University of Glasgow Centre for Virus Research, Glasgow, UK.

[4]Stanford University School of Medicine, Stanford, California, USA. [5]Ragon Institute of MGH, MIT, and Harvard, Cambridge, Massachusetts 02139, USA. [6]Ph.D. Program in Virology, Division of Medical Sciences, Harvard Medical School, Boston, Massachusetts 02115, USA. [7]Computational and Systems Biology, Massachusetts Institute of Technology, Cambridge, Massachusetts, USA. [8]Integrated Research Facility, Division of Clinical Research, National Institute of Allergy and Infectious Diseases, National Institutes of Health, Frederick, Maryland, USA. [9]Center for Genome Sciences, The United States Army Medical Research Institute for Infectious Disease, 1425 Porter Street, Fort Detrick, Maryland 21702, USA. [10]Center for Systems Biology, Department of Organismic and Evolutionary Biology, Harvard University, Cambridge, Massachusetts, USA. [11]Division of Infectious Diseases, Beth Israel Deaconess Medical Center, Boston, Massachusetts, USA. [12]African Center of Excellence for Genomics of Infectious Disease (ACEGID), Redeemer's University, Ede, Osun State, Nigeria. [13]Department of Biological Sciences, College of Natural Sciences, Redeemer's University, Ede, Osun State, Nigeria. [14]Institute of Lassa Fever Research and Control, Irrua Specialist Teaching Hospital, Irrua, Edo State, Nigeria. [15]Viral Hemorrhagic Fever Program, Kenema Government Hospital, Kenema, Sierra Leone. [16]Department of Electrical Engineering and Computer Science, MIT, Cambridge, Massachusetts 02139, USA. [17]Zalgen Labs, Germantown, Maryland, USA. [18]Department of Immunology and Microbial Science, The Scripps Research Institute, La Jolla, California, USA. [19]Emerging Infectious Diseases Branch, Walter Reed Army Institute of Research, Silver Spring, Maryland, USA. [20]Henry M. Jackson Foundation for the Advancement of Military Medicine, Bethesda, Maryland, USA. [21]Henry M. Jackson Foundation Medical Research International, Abuja, Nigeria. [22]Ministry of Health and Sanitation, Freetown, Sierra Leone. [23]Tulane University School of Medicine, New Orleans, Los Angeles 70112, USA. [24]Howard Hughes Medical Institute, Chevy Chase, Maryland, USA. [25]These authors contributed equally: Kayla G. Barnes, Anna E. Lachenauer. ✉email: kbarnes@broadinstitute.org; annalach@stanford.edu

