## [Peer Review File · Nature Communications]

Reviewers' Comments:

Reviewer #1:

Remarks to the Author:

Barnes et. al. provide data supporting the implementation of a novel CRISPR based technology to the diagnostics of two high priority viral hemorrhagic fevers, LASV and EBOV, which have proven an incredible public health burden through out Western and Central Africa. The work is timely and attempts to address many issues that workers in these otherwise austere clinical settings are faced with in terms of utilizing cutting edge, or even basic, diagnostic services for high consequence emerging and endemic viral diseases. The study design and supporting statistical analysis appear adequate. A few issues that should be clarified or addressed prior to publication include.

1. Line 94 and throughout text: References used, are these really the "gold standard" assays? Are there not commercial kits that target LASV? How does this statement fit in with WHO or US-CDC (or Africa CDC, or any CDC for that matter) guidance on LASV detection? Are these assays what MSF or other academic groups are using? I'm not sure there is a "gold standard" unless you are talking about a methodology ie RT-PCR.

2. Hudson heating inactivation: This is a pretty big safety concern, I strongly recommend a statement about minimum biosafety conditions required to do this inactivation. I.E. use of a BSC, glove box, PPE requirements, screw cap or snap cap tube (the latter is a huge biosafety aerosol risk as snap cap tubes tend to pop when heated), any vortexing? Please provide more detail on the process as this manuscript will be looked at by folks attempting to replicate what you have done and providing a very firm and serious tone to biosafety here will be appreciated by all who read and evaluate your work.

3. Line 133 and throughout text where any incidence of the word "isolate" is used. The authors are using this term way too loosely. An isolate, in the classical sense, refers to an "isolated virus". A clinical sample could have a "swarm" of different "isolates" within it. So when you refer to a type of sample it is important to be mindful of what the implications are here. From this reviewer's perspective, you were testing clinical samples, not isolates. Please review the rest of the manuscript for use of "isolates" and correct accordingly.

4. Line 148-149, change sample storage to cold chain.

5. I see there is use of full length viral sequences of EBOV that were used? Is this correct or did I misread something? The use of full length genomes of select agents where transcription or other biological processes is pretty tightly regulated by NIH-RAC. Can the authors confirm these experiments are exempt from NIH-RAC regulation? I understand that no attempts to produce infectious virus were made, but the use of full length EBOV genomes is creeping into a grey area and I would hate to have the authors get this published and not be in compliance with perhaps an otherwise not realized regulation. Please address this important issue.

Reviewer #2:

Remarks to the Author:

As the next-generation diagnostics, CRISPR-Dx has the merits of high sensitivity, much convenience, rapidness, etc. Among the CRISPR-Dx systems, Cas13-based and Cas12-based systems are the most promising. Through combing HUDSON, which provides a convenient solution for clinical sample preparation, and Cas13-based SHERLOCK system, the authors here developed deployable diagnostic tools for realtime detection of Ebola and Lassa viruses. As expected, the tool is of high specificity and sensitivity. Besides, with the assistance of HandLens, the results can be read out on mobile phones. Overall, the work is of importance, especially at the special moment. I only have several minor points.

1. How to avoid aerosol contamination during the operations.

2. During the whole processes, pipettes and incubators are still required, which is inconvenient and could restrict the technology from field application. The authors may discuss about this and find out a solution.

3. Besides of SHERLOCK, there are also Cas12-based CRISPR-Dx. Can the authors explain the

rationale of choosing SHERLOCK in detection of VHF viruses?

4. As the ID NOW technology from Abbott has recently been applied in SARS-CoV-2 detection, which is extremely fast, the authors should compare their technology with ID now.

Dear Reviewers,

Thank you for very much for your thoughtful and positive view of our manuscript NCOMMS-20-11096A. We are pleased to see that both referee's appreciated the impact and timeliness of the paper. We have been able to respond to each constructive and specific comment. Below are our point-by-point responses.

Thank you for your consideration and continued efforts.

Sincerely,

Reviewer #1 (Remarks to the Author):

Barnes et. al. provide data supporting the implementation of a novel CRISPR based technology to the diagnostics of two high priority viral hemorrhagic fevers, LASV and EBOV, which have proven an incredible public health burden throughout Western and Central Africa. The work is timely and attempts to address many issues that workers in these otherwise austere clinical settings are faced with in terms of utilizing cutting edge, or even basic, diagnostic services for high consequence emerging and endemic viral diseases. The study design and supporting statistical analysis appear adequate. A few issues that should be clarified or addressed prior to publication include.

1. Line 94 and throughout text: References used, are these really the “gold standard” assays? Are there not commercial kits that target LASV? How does this statement fit in with WHO or US-CDC (or Africa CDC, or any CDC for that matter) guidance on LASV detection? Are these assays what MSF or other academic groups are using? I’m not sure there is a “gold standard” unless you are talking about a methodology ie RT-PCR.	We have changed “gold-standard” to “most widely used” diagnostic on line 98 and deleted gold-standard elsewhere in the manuscript. Unfortunately, there has been very little development for Lassa fever diagnostics over the last decade. Until 2018, to our knowledge, the few centers that routinely tested (Kenema Government Hospital (KGH), Sierra Leone and Irrua Specialist Teaching Hospital (ISTH), Nigeria) used Nikisin et al. primer sets. All testing in Sierra Leone are still carried out by KGH. During the 2018 Lassa Fever outbreak in Nigeria more testing centers were set up by the NCDC but ISTH remained the main testing center and continued to use Nikisin et al primer set. One academic group did use the new commercially available (Altona kit) which included the Nikisin et al primers. Currently this kit is not widely available or affordable compared to directly purchasing the Nikisin et al. primers. To our knowledge MSF and other organizations outsource testing to the centers mentioned above.
2. Hudson heating inactivation: This is a pretty big safety concern, I strongly recommend a statement about minimum biosafety conditions required to do this inactivation. I.E. use of a BSC, glove box, PPE requirements, screw cap or snap cap tube (the latter is a huge biosafety aerosol risk as snap cap tubes tend to pop when heated), any vortexing? Please provide more detail on the process as this manuscript will be looked at by folks attempting to replicate what you have done and providing a very firm and serious tone to biosafety here will be appreciated by all who read and evaluate your work.	Yes, this is an important point. We spent a lot of time ensuring the safety of HUDSON in BL4 facilities so thank you for flagging this. We have added safety requirements in main text lines 227-228 We have also added them to the methods: lines 289-290 lines 406-410 lines 422-424

3. Line 133 and throughout text where any incidence of the word “isolate” is used. The authors are using this term way too loosely. An isolate, in the classical sense, refers to an “isolated virus”. A clinical sample could have a “swarm” of different “isolates” within it. So when you refer to a type of sample it is important to be mindful of what the implications are here. From this reviewer’s perspective, you were testing clinical samples, not isolates. Please review the rest of the manuscript for use of “isolates” and correct accordingly.	This is a very fair point. We have removed isolate from line 140, and from any place where we are looking at direct work with clinical samples. For lines 159-167 these are one unique isolate derived from a clinical sample so it fits with the definition.
4. Line 148-149, change sample storage to cold chain.	We have corrected this (now on lines 156-157)
5. I see there is use of full length viral sequences of EBOV that were used? Is this correct or did I misread something? The use of full length genomes of select agents where transcription or other biological processes is pretty tightly regulated by NIH-RAC. Can the authors confirm these experiments are exempt from NIH-RAC regulation? I understand that no attempts to produce infectious virus were made, but the use of full length EBOV genomes is creeping into a grey area and I would hate to have the authors get this published and not be in compliance with perhaps an otherwise not realized regulation. Please address this important issue.	In this paper there are two uses of synthetically derived DNA.  1. Cross reactivity and LOD. We have clarified in the methods that only a small non-replicant competent fragment was used on Line 348 and lines 358-360 2. Testing of DRC isolate - Lines 159-167: Yes, the work carried out on the DRC sample and Makona isolate are full length genomes. These synthetic isolates where made at USAMRIID under BL4 conditions and under full regulatory approval. For this work we sent SHERLOCK reagents and the 1-page instructions to USAMRIID and Gustavo Palacios’s team (specifically Brett Beitzel) carried out the testing. We have clarified the work that occurred at USAMRIID under their regulatory approvals in methods Line 393-394

Reviewer #2 Comments to Authors

As the next-generation diagnostics, CRISPR-Dx has the merits of high sensitivity, much convenience, rapidness, etc. Among the CRISPR-Dx systems, Cas13-based and Cas12-based systems are the most promising. Through combining HUDSON, which provides a convenient solution for clinical sample preparation, and Cas13-based SHERLOCK system, the authors here developed deployable diagnostic tools for realtime detection of Ebola and Lassa viruses. As expected, the tool is of high specificity and sensitivity. Besides, with the assistance of HandLens, the results can be read out on mobile phones. Overall, the work is of importance, especially at the special moment. I only have several minor points.

1. How to avoid aerosol contamination during the operations.	Yes, this is an important point. We spent a lot of time ensuring the safety of HUDSON in BL4 facilities so thank you for flagging this. With these BL4 pathogens aerosols can quickly infect a HCW so proper PPE is imperative. We have added safety requirements in main text lines 227-228 We have also added them to the methods: lines 289-290 lines 406-410 lines 422-424
2. During the whole processes, pipettes and incubators are still required, which is inconvenient and could restrict the technology from field application. The authors may discuss about this and find out a solution.	Yes, the reviewer is correct that minimal laboratory equipment is required and although we are working to improve our 1-pot system that work is not part of this manuscript. For this work we only require a heat blocks and pipette – which we have clarified on line 109. This minimal equipment can be used in any environment and from our experience by any user including field workers with limited training.
3. Besides of SHERLOCK, there are also Cas12-based CRISPR-Dx. Can the authors explain the rationale of choosing SHERLOCK in detection of VHF viruses?	Cas13 (formally referred to as C2C2) targets RNA while Cas12 targets DNA. Since Lassa, Ebola and other key viral pathogens are RNA viruses Cas13 can recognize the viral RNA target and upon recognition stare a collateral cleavage allowing the detection we describe in this paper and previously (See Gootenberg et al. Science 2017 for a full description).
4. As the ID NOW technology from Abbott has recently been applied in SARS-CoV-2 detection, which is extremely fast, the authors should compare their technology with ID now.	Currently the Abbott does not have a Lassa fever or Ebola virus test. Therefore, we cannot do a direct comparison. However, we recognize that there will be interest in this and other technologies given the COVID-19 pandemic. We have added an additional discussion (lines 256-261) in our final paragraph to highlight the changing landscape of diagnostics and how our technology still remains cheaper and requires less overall machinery.

Reviewers' Comments:

Reviewer #1:

Remarks to the Author:

All points have been satisfactorily addressed.

Dear Reviewers,

Thank you for very much for your thoughtful and positive view of our manuscript NCOMMS-20-11096A. We are pleased to see that both referee's appreciated the impact and timeliness of the paper. We have been able to respond to each constructive and specific comment. Below are our point-by-point responses.

Thank you for your consideration and continued efforts.

Sincerely,

Reviewer #1 (Remarks to the Author):

Barnes et. al. provide data supporting the implementation of a novel CRISPR based technology to the diagnostics of two high priority viral hemorrhagic fevers, LASV and EBOV, which have proven an incredible public health burden throughout Western and Central Africa. The work is timely and attempts to address many issues that workers in these otherwise austere clinical settings are faced with in terms of utilizing cutting edge, or even basic, diagnostic services for high consequence emerging and endemic viral diseases. The study design and supporting statistical analysis appear adequate. A few issues that should be clarified or addressed prior to publication include.

1. Line 94 and throughout text: References used, are these really the "gold standard" assays? Are there not commercial kits that target LASV? How does this statement fit in with WHO or US-CDC (or Africa CDC, or any CDC for that matter) guidance on LASV detection? Are these assays what MSF or other academic groups are using? I'm not sure there is a "gold standard" unless you are talking about a methodology ie RT-PCR.

We have changed "gold-standard" to "most widely used" diagnostic on line 98 and deleted gold-standard elsewhere in the manuscript.

Unfortunately, there has been very little development for Lassa fever diagnostics over the last decade. Until 2018, to our knowledge, the few centers that routinely tested (Kenema Government Hospital (KGH), Sierra Leone and Irrua Specialist Teaching Hospital (ISTH), Nigeria) used Nikisin et al. primer sets. All testing in Sierra Leone are still carried out by KGH. During the 2018 Lassa Fever outbreak in Nigeria more testing centers were set up by the NCDC but ISTH remained the main testing center and continued to use Nikisin et al primer set. One academic group did use the new commercially available (Altona kit) which included the Nikisin et al primers. Currently this kit is not widely available or affordable compared to directly purchasing the Nikisin et al. primers.

To our knowledge MSF and other organizations outsource testing to the centers mentioned above.

2. Hudson heating inactivation: This is a pretty big safety concern, I strongly recommend a statement about minimum biosafety conditions required to do this inactivation. I.E. use of a BSC, glove box, PPE requirements, screw cap or snap cap tube (the latter is a huge biosafety aerosol risk as snap cap tubes tend to pop when heated), any vortexing? Please provide more detail on the process as this manuscript will be looked at by folks attempting to replicate what you have done and providing a very firm and serious tone to biosafety here will be appreciated by all who read and evaluate your work.	Yes, this is an important point. We spent a lot of time ensuring the safety of HUDSON in BL4 facilities so thank you for flagging this. We have added safety requirements in main text lines 227-228 We have also added them to the methods: lines 289-290 lines 406-410 lines 422-424
3. Line 133 and throughout text where any incidence of the word “isolate” is used. The authors are using this term way too loosely. An isolate, in the classical sense, refers to an “isolated virus”. A clinical sample could have a “swarm” of different “isolates” within it. So when you refer to a type of sample it is important to be mindful of what the implications are here. From this reviewer’s perspective, you were testing clinical samples, not isolates. Please review the rest of the manuscript for use of “isolates” and correct accordingly.	This is a very fair point. We have removed isolate from line 140, and from any place where we are looking at direct work with clinical samples. For lines 159-167 these are one unique isolate derived from a clinical sample so it fits with the definition.
4. Line 148-149, change sample storage to cold chain.	We have corrected this (now on lines 156-157)
5. I see there is use of full length viral sequences of EBOV that were used? Is this correct or did I misread something? The use of full length genomes of select agents where transcription or other biological processes is pretty tightly regulated by NIH-RAC. Can the authors confirm these experiments are exempt from NIH-RAC regulation? I understand that no attempts to produce infectious virus were made, but the use of full length EBOV genomes is creeping into a grey area and I would hate to have the authors get this published and not be in compliance with perhaps an otherwise not realized regulation. Please address this important issue.	In this paper there are two uses of synthetically derived DNA.  1. Cross reactivity and LOD. We have clarified in the methods that only a small non-replicant competent fragment was used on Line 348 and lines 358-360 2. Testing of DRC isolate - Lines 159-167: Yes, the work carried out on the DRC sample and Makona isolate are full length genomes. These synthetic isolates were made at USAMRIID under BL4 conditions and under full regulatory approval. For this work we sent SHERLOCK reagents and the 1-page instructions to USAMRIID and Gustavo Palacios’s team (specifically Brett Beitzel) carried out the testing. We have clarified the work that

	occurred at USAMRIID under their regulatory approvals in methods Line 393-394
--	---

Reviewer #2 Comments to Authors

As the next-generation diagnostics, CRISPR-Dx has the merits of high sensitivity, much convenience, rapidness, etc. Among the CRISPR-Dx systems, Cas13-based and Cas12-based systems are the most promising. Through combing HUDSON, which provides a convenient solution for clinical sample preparation, and Cas13-based SHERLOCK system, the authors here developed deployable diagnostic tools for realtime detection of Ebola and Lassa viruses. As expected, the tool is of high specificity and sensitivity. Besides, with the assistance of HandLens, the results can be read out on mobile phones. Overall, the work is of importance, especially at the special moment. I only have several minor points.

1. How to avoid aerosol contamination during the operations.	Yes, this is an important point. We spent a lot of time ensuring the safety of HUDSON in BL4 facilities so thank you for flagging this. With these BL4 pathogens aerosols can quickly infect a HCW so proper PPE is imperative. We have added safety requirements in main text lines 227-228 We have also added them to the methods: lines 289-290 lines 406-410 lines 422-424
2. During the whole processes, pipettes and incubators are still required, which is inconvenient and could restrict the technology from field application. The authors may discuss about this and find out a solution.	Yes, the reviewer is correct that minimal laboratory equipment is required and although we are working to improve our 1-pot system that work is not part of this manuscript. For this work we only require a heat blocks and pipette – which we have clarified on line 109. This minimal equipment can be used in any environment and from our experience by any user including field workers with limited training.
3. Besides of SHERLOCK, there are also Cas12-based CRISPR-Dx. Can the authors explain the rationale of choosing SHERLOCK in detection of VHF viruses?	Cas13 (formally referred to as C2C2) targets RNA while Cas12 targets DNA. Since Lassa, Ebola and other key viral pathogens are RNA viruses Cas13 can recognize the viral RNA target and upon recognition stare a collateral cleavage allowing the detection we describe in this paper and previously (See Gootenberg et al. Science 2017 for a full description).
4. As the ID NOW technology from Abbott has recently been applied in SARS-CoV-2 detection, which is extremely fast, the authors should compare their technology with ID now.	Currently the Abbott does not have a Lassa fever or Ebola virus test. Therefore, we cannot do a direct comparison. However, we recognize that there will be interest in this and other technologies given the COVID-19 pandemic. We have added an additional discussion

	(lines 256-261) in our final paragraph to highlight the changing landscape of diagnostics and how our technology still remains cheaper and requires less overall machinery.
--	--